# Study on Ultrasonically-Enhanced Deep Eutectic Solvents Leaching of Zinc from Zinc-Containing Metallurgical Dust Sludge

**Fusheng Niu** [1,2,*]**, Shengtao He** [1]**, Jinxia Zhang** [1,2,*] **and Chen Wen** [1,2]

[1] College of Mining Engineering, North China University of Science and Technology, Tangshan 063009, China
[2] Hebei Province mining industry develops with safe technology priority laboratory, Tangshan 063009, China
[*] Correspondence: niufusheng@ncst.edu.cn (F.N.); zhangjinxia163@163.com (J.Z.)

**Abstract:** In this study, the zinc containing dust and sludge of iron and steel smelting was taken as the research object, and the new ionic liquid of choline-urea was prepared and synthesized as the leaching agent. The conventional and ultrasonic leaching of zinc were compared, and the influence of liquid-solid ratio, temperature, time, ultrasonic power and other conditions on the zinc leaching rate were analyzed. The leaching residue was characterized by Scanning Electron Microscopy (SEM), Energy Dispersive Spectroscopy (EDS), X-ray diffraction (XRD), and the kinetic equations of ligand leaching based on ultrasonically enhanced metallurgical dust sludge were constructed. The results showed that the choline chloride-urea ionic liquid has a special solubilization ability for ZnO, and the leaching rate of Zn at temperature 60 °C, ultrasonic power 350 W, and leaching time 240 min reached more than 98%. Kinetic fitting of the ChCl-urea leaching process revealed that the ChCl-urea leaching process was in accordance with the nucleation contraction model under both conventional and ultrasonic conditions, and the leaching process was dominated by interfacial mass transfer and solid film layer diffusion control for the reactions, respectively. The activation energies were $Ea_1 = 44.56$ kJ/mol and $Ea_2 = 23.06$ kJ/mol

**Keywords:** leaching; zinc-containing dust sludge; ultrasound; kinetics

## 1. Introduction

As an important basic resource of non-ferrous metals, zinc is widely used in automobiles, construction, shipbuilding, the light industry and other industries, and is a supporting material for industrial development, a key material for national defense construction and a basic material for economic strategic development, and its supply guarantee is an important foundation for the sustainable development of the national economy [1]. According to the National Bureau of Statistics, in 2021, China's zinc output will be 6.56 million tons. Although China's zinc production still ranks first in the world, the supply gap of zinc resources is still large due to the large number of low-grade ores and few rich ores in primary resources, resulting in China's high external dependence on zinc [2,3].

Metallurgical dust sludge is a solid waste discharged during the production of iron and steel industry, which is rich in valuable components such as iron, carbon, zinc and lead and reusable alkali metal substances such as calcium and magnesium, therefore, metallurgical dust sludge is a resource containing a large number of valuable components that can be utilized in depth and comprehensively [4,5]. However, the traditional disposal method is mostly to return directly to sintering, which will not only affect the normal operation of the blast furnace due to the accumulation of non-ferrous substances, but also increase the emission of volatile and highly toxic pollutants such as lead, zinc, cadmium, antimony and bismuth into the environment [6,7].

With the depletion of primary zinc ore resources, industrial solid wastes such as metallurgical dust sludge and zinc smelting slag have become an important source of recycled zinc raw materials in China, which is an important way to solve the shortage of resources in China if suitable technologies are adopted to recover valuable metals such as zinc and iron from them [3,8,9]. Therefore, the efficient resource utilization of metallurgical dust sludge has become a hot issue for the industry to solve urgently, and is also an important area and main direction for the development of circular economy and improvement of comprehensive utilization of resources in the steel industry in the coming period.

Zhang Jinxia et al. [9] investigated the effects of sulfuric acid concentration, liquid-to-solid ratio, stirring speed and reaction temperature on the leaching rate of zinc by acid leaching. The results showed that under the conditions of sulfuric acid concentration of 0.9 mol/L, liquid-solid ratio of 6:1 (mL:g), stirring speed of 300 r/min, and reaction time of 40 min, the Zn leaching rate reached 96.30%. In traditional acid leaching treatment, there is a large amount of acid consumption and it is easy to enter the leaching solution of iron, resulting in the subsequent iron removal process being complex; using the ammonia leaching process developed in recent years, and the difficult-to-dissolve zinc ferrite, there are problems of a low leaching rate, ammonia volatility, leaching process-leaching agent diffusion, slow speed and other problems [10,11]. In particular, the secondary pollution caused by the recovery of useful minerals from solid waste also limits the application of this technology [12,13]. Therefore, there is a need to develop a new method with mild conditions, low environmental footprint and high efficiency.

Deep eutectic solvents (DESs) [14,15], first proposed by Abbott, Capper and Davies et al. were used to dissolve metal oxides. The principle is a two- or three-component low eutectic mixture consisting of a certain stoichiometric ratio of hydrogen-bonded acceptor and hydrogen-bonded donor, with different solubility of different metal oxides in different low eutectic solvents. Bing Lu et al. [16] used DES composed of choline chloride （CHClNO）and malonic acid（C3H4O4）for the leaching of waste lithium-ion cathode materials by exploiting the property of DES to dissolve oxides, and the leaching efficiency of cobalt and lithium reached 98.61% and 98.78%, respectively, at a lower leaching temperature (100 °C). Mehmet A et al. [17] used ChCl-urea DES as leaching agent and determined the optimum leaching conditions for the recovery of precious metals from anode sludge using Taguchi method. $L_{16}(4^4)$ orthogonal experimental design was used to study the leaching parameters such as DES components, reaction temperature, reaction time, and material-to-liquid ratio, and the leaching yield of copper was as high as 97% at a leaching temperature of 95 °C, a leaching time of 4 h, and a solid-to-liquid ratio of 1:25g/ml. The leaching behavior of chalcopyrite in the ionic liquid [bmim]HSO4 was studied systematically by Tie-Guang Dong, and the leaching rate of copper was increased by nearly 3–4 times compared with the leaching of chalcopyrite in the conventional $H_2SO_4$-$Fe_2(SO_4)_3$ system [14].

Since the discovery of the chemical effect of ultrasound by W.T. Richards and A.L. Loomis in 1923, the application of ultrasonic enhancement in wet leaching processes has gradually received attention [18–20]. The cavitation phenomenon occurs when the ultrasonic wave with high enough energy acts on the liquid, and the impact effect, pressure effect and micro-jet effect shown by the cavitation effect of ultrasound can form surface etchings and boundary layer cavities on the boundary layer and particle surface, reducing the thickness of the boundary layer and increasing the interfacial reaction area, effectively reducing the adverse effects such as covering and wrapping of insoluble minerals [21,22]. At the same time, ultrasonic action can broaden the chemical reaction channels, accelerate the speed of chemical reactions and strengthen the process of material transfer [23,24]. Yang et al. [25,26] enhanced the leaching of ZnO ore by using ultrasonic external field enhancement technique and ammonium citrate ligand leaching agent. The study showed that the ammonium citrate ligand leaching process could form stable complexes, and the zinc leaching rate of ultrasonic-ammonium citrate ligand enhanced ammonia-ammonium chloride system increased by 36.7% compared with the better leaching rate of the original

ore. Yanfang Zhang et al [27] studied the acid leaching kinetics of potassium feldspar with sulfuric acid under ultrasonic system, and the results showed that due to ultrasonic cavitation, ultrasonic waves with sulfuric acid as leaching agent could enhance the acid leaching process of potassium feldspar, and the potassium leaching rate could be increased by 3% to 8%.

In this paper, metallurgical zinc-containing dust sludge was used as raw material, and the ultrasonic method was used to supplement a ChCl-urea DESs leaching process. We investigate the effects of liquid-solid ratio, leaching temperature, leaching time and ultrasonic power on the leaching rate of zinc in the leaching reaction and to obtain the best process for leaching zinc. The leaching kinetics of ChCl-urea DESs were studied in the ultrasonic regime to determine the ultrasonically enhanced metallurgical dust sludge coordination leaching kinetic model and to calculate the apparent activation energy.

## 2. Materials and Methods

### 2.1. Raw Materials

In this study, zinc-containing dust sludge generated from the steel smelting process of a local steel plant in Shanxi was used as raw material. Figures 1 and 2 show the main composition and microstructure of the samples. From the X-ray diffraction (XRD-D8 Advance, Bruker, Karlsruhe, Germany) results in Figure 1, it can be seen that the sample is complex in composition, with the main components being iron-bearing minerals such as hematite and magnetite, and Zn mainly present in zinc chloride hydrate ($Zn_5(OH)_8Cl_2$), red zincite (ZnO), sphalerite (ZnS) and zinc ferrite ($ZnFe_2O_4$).

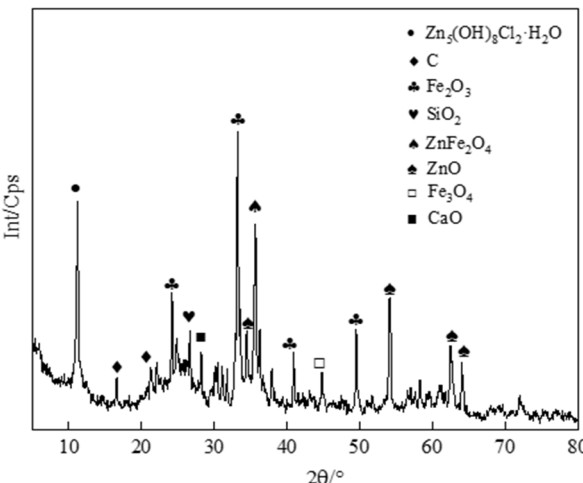

**Figure 1.** X-ray diffraction pattern of the zinc-containing dust.

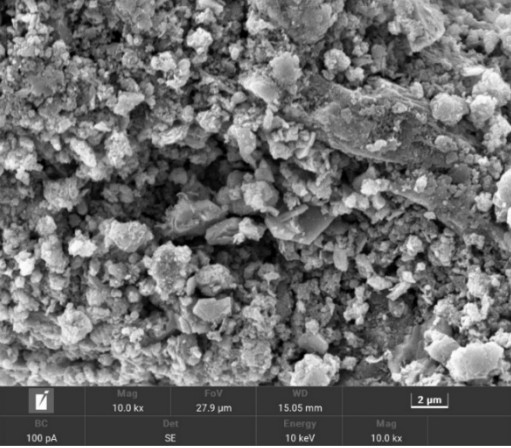

**Figure 2.** SEM of the zinc-containing dust.

Table 1 shows the main chemical composition of this Zn-bearing dust sludge sample. The main elements in the dust sludge are Fe, Zn and Si, containing 11.78% ZnO, 30.67% Fe2O₃ and 5.98% SiO₂.

**Table 1.** Chemical multielement analysis of zinc-bearing dust sludge (%).

| Element | $Fe_2O_3$ | ZnO | $SiO_2$ | Cl | $Al_2O_3$ | CaO | PbO | MgO | $K_2O$ |
|---|---|---|---|---|---|---|---|---|---|
| content/% | 30.67 | 11.78 | 5.98 | 5.32 | 3.657 | 3.253 | 1.931 | 2.125 | 1.203 |

The results of the laser particle size analysis of the Zn-containing dust samples are shown in Figure 3. It can be seen that the volume weighted average particle size of the Zn-containing dust samples is 75.1 um.

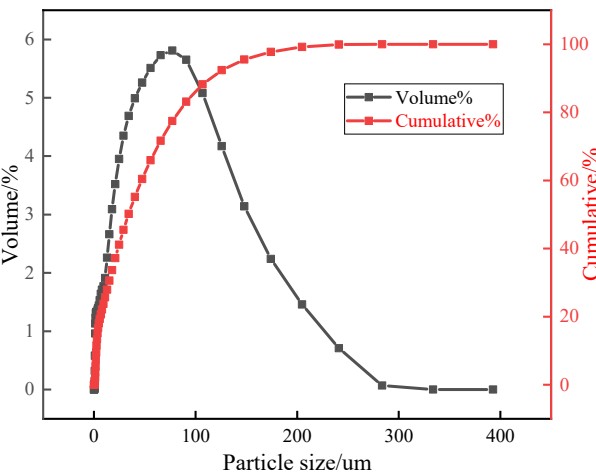

**Figure 3.** Laser particle size results of zinc-bearing dust samples.

## 2.2. Experimental Methods and Devices

Selection of the molar ratio of ChCl to urea: the relevant literature shows that the melting point is a prerequisite for DESs to be widely used, and the lower the melting point of the formed DESs, the more stable its physical and chemical properties. The melting points of the mixtures of ChCl and urea at different molar ratios at room temperature (25 °C) were measured and the results are shown as follows in Figure 3.

As can be seen from Figure 4, the melting points of the mixtures both decrease and then increase with the increase in the amounts of ChCl and urea. The lowest melting point of the mixture was obtained when ChCl was mixed with urea in a molar ratio of 1:2. Therefore, the molar ratio of the low eutectic solvent chosen for the subsequent experiments was ChCl:urea=1:2.

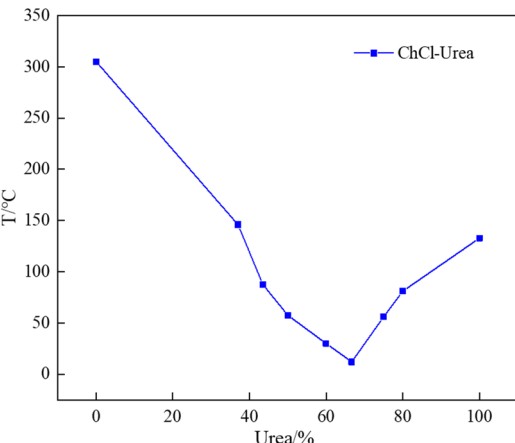

**Figure 4.** Melting point of ChCl-urea mixture.

Preparation of deep eutectic solvents: a certain amount of choline chloride (ChCl, analytical pure, Chinese medicine) and urea (urea, analytical pure, Sinopharm Chemical Reagent Co.,Ltd) were dried in a vacuum drying oven, with setting parameters as follows: 80 °C, −0.05 MPa, 10 h. The dried choline chloride and urea were mixed thoroughly in a 1 L beaker in a molar ratio of 1:2, and then placed in a vacuum drying oven, with setting parameters as follows: 80 °C, −0.05 MPa, 12 h, until a transparent homogeneous liquid is obtained, sealed and reserved.

Conventional leaching: leaching experiments were carried out in 250 mL beakers. First, 10 g of Zn-containing dust sludge was taken in a 250 mL round beaker for each experiment, and prepared ChCl-urea DESs were added at different liquid-to-solid ratios, and then the beaker was placed on a hexagonal thermoregulated magnetic stirrer equipped with a water bath. The leaching temperature and stirring rate were controlled using a sextuplex asynchronous electric magnetic stirrer and the leaching time was recorded.

Ultrasonic leaching: during the leaching process, the ultrasonic probe was placed at 1.5–2 cm below the liquid surface except. The details of the treatment process were similar to those described above for conventional leaching, except that the magnetic stirring in conventional leaching was replaced by different ultrasonic powers.

Characterization: in order to determine the leaching rate of Zn, the leaching liquor was separated from the leaching residue after the experiment by using a vacuum filtration machine (XTLZ-φ260/φ200, Shaoxing Tianyun Instrument & Equipment Co. Ltd., Shaoxing, China), and the filtrate was dried and weighed. Some of the filtrate was dissolved and the zinc content was measured by atomic absorption spectrophotometer, and the leaching rate of zinc was calculated. The zinc content of the leachate after leaching was analyzed by atomic absorption spectrometry (AA-3600, Shimadzu, Kyoto, Japan). SEM-EDS (SPM-S3400N, Hitachi, Tokyo, Japan) and x-ray powder diffraction (XRD) were used to analyze the microstructure and phase changes of the Zn-containing dust sludge before and after leaching.

Experimental setup: the experimental setup is shown in Figure 5. The experimental setup included an ultrasonic device consisting of an ultrasonic generator (FS-900 N, Shanghai Sonxi Ultrasonic Instrument Co., Ltd, Shanghai, China) and a titanium alloy ultrasonic probe with a diameter of 6 mm in addition to the conventional leaching device.

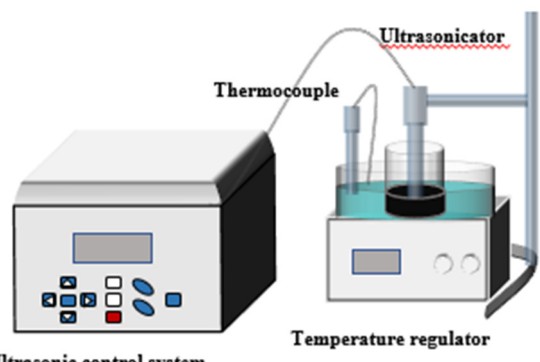

**Figure 5.** The ultrasonically enhanced leaching system.

### 2.3. Leaching Mechanism and Mathematical model

The study of the leaching mechanism from the point of view of the leaching kinetics will help to determine the control steps of the reaction process and to carry out targeted step enhancement [28]. The ChCl-2urea and ZnO(001) surfaces were modeled using the software Material Studio (Material Studio 2019, Accelrys Software Inc, San Diego, CA, USA), and the interaction model was obtained as shown in Figure 4. It can be seen from Figure 6 that for ChCl-2urea, two low eutectic solvents interact on the upper and lower surfaces of ZnO(001). The ChCl-2ureaDES combined through multiple hydrogen bonds decomposes during the interaction with the ZnO(001) surface. In which the Cl in choline chloride and the hydrogen bond donor urea act on the Zn protruding surface, the choline cation in choline chloride acts on the O protruding surface in the ZnO(001) surface, and part of the C-H bond in the choline cation forms multiple hydrogen bonds with the O on the ZnO surface to adsorb on the ZnO surface, eventually producing the complex $[ZnOCl(NH_2CONH_2)_2]^-$, thus dissolving zinc oxide.

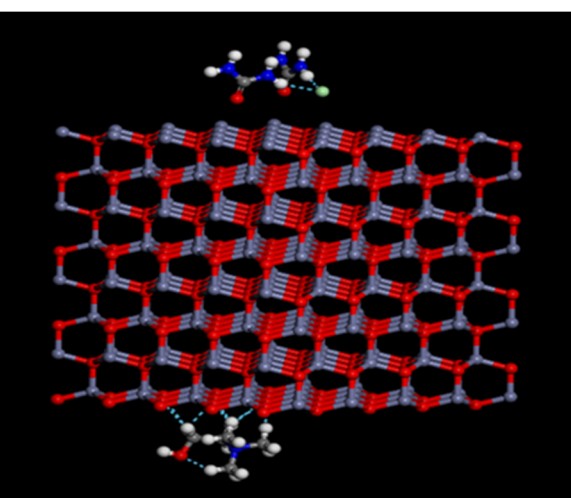

**Figure 6.** The adsorption structure of ZnO and ChCl-2Urea DES.

The possible leaching mechanism of ZnO in ChCl-2ureaDES is,

$$ZnO + 2(NH_2CONH_2) + [Cl]^- \leftrightarrow [ZnOCl(NH_2CONH_2)_2]^- \qquad (1)$$

The leaching process is shown in Figure 7, and the reactants are uniform powder particles, which can be regarded as spheres. The starting radius of the Zn-containing dust particles is $r_0$, and with the gradual depth of the leaching process, the reaction interface keeps shrinking toward the nucleus, and after time t, the radius of the unreacted nucleus is rt. The residual solid layer that does not participate in the reaction wraps the unreacted

nucleus, and the leaching agent keeps penetrating inward and [ZnOClHOOCCH2COOH]⁻ keeps diffusing outward, and finally the remaining quartz and residues such as hematite are left. The reaction between ZnO and ChCl-urea DES in Zn-containing dust sludge is a solid-liquid two-phase reaction, and the shape of the particles remains basically unchanged during the leaching process, which is suitable for the kinetic analysis of the leaching reaction using the nuclear contraction model, and the changes of its leaching rate may be controlled by a variety of systems, mainly including those controlled by liquid film diffusion, solid film layer diffusion, the slowest process in the surface chemical reaction, or the mixing between them.

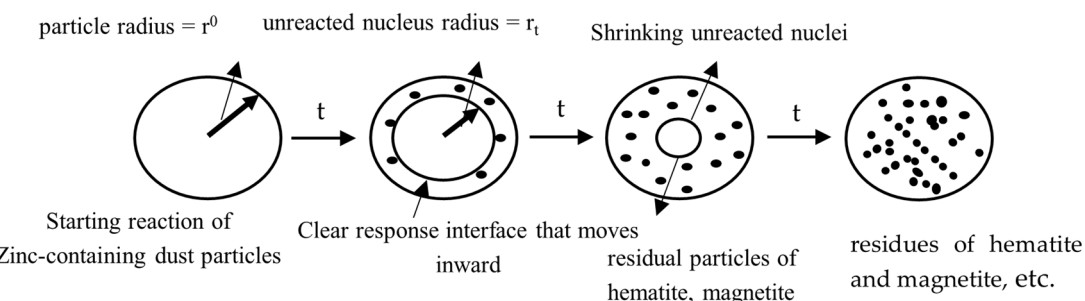

**Figure 7.** Schematic diagram of the leaching process of zinc-containing dust sludge particles.

When the shape of the solid particles during leaching is nearly spherical and controlled by the diffusion of the residual solid film layer, the following kinetic model is generally fitted [29–31].

$$1 - 2/3x - (1-x)^{2/3} = k_d t \qquad (2)$$

Where:
$x$—leaching rate of ZnO in Zn-containing dust sludge at moment t.
$k_d$—the rate constant of diffusion control, min⁻¹.
$t$—is the leaching time, min.

If the reaction is controlled by chemical reactions, the leaching kinetic equation of the contraction Kernel model can be expressed as:

$$1 - 2/3x - (1-x)^{2/3} = k_r t \qquad (3)$$

Where:
$x$—leaching rate of ZnO from Zn-containing dust sludge at moment t.
$k_r$—the rate constant of chemical control, min⁻¹.
$t$—is the leaching time, min.

No solid product layer is generated during the leaching of Zn-containing dust in ChCl-2urea, but because of the complex composition of Zn-containing dust and the possibility of non-reactive substances such as quartz adhering to the surface of the reaction particles during the reaction, solid product layer diffusion is also applicable to this reaction, and a suitable reaction model needs to be determined with comprehensive consideration. Bingöl D. et al. [32] applied a novel kinetic model to explain this process based on interfacial mass transfer and solid film layer diffusion using a novel kinetic model with the following Equations.

$$1/3\ln(1-x) - 1 + (1-x)^{-1/3} = kt \qquad (4)$$

Where,
$x$—leaching rate of ZnO in Zn-containing dust sludge at moment t.
k—reaction rate constant, min⁻¹.

$t$—leaching time, min.

The calculation of the leaching rate of Zn can be expressed as:

$$\varepsilon = \frac{V_1 \times c_1}{m_0 \times x_0} \times 100\% \tag{5}$$

Where,

$m_0$—mass of the zinc-containing dust sludge sample, g.

$x_0$—the zinc content of the zinc-containing dust sludge sample, %.

$V_1$—is the volume of leachate, L.

$c_1$—content of zinc in the leaching solution, g.L$^{-1}$.

## 3. Results and Discussion

### 3.1. Effect of Leaching Temperature

From Figure 8a, it can be seen that the leaching rate of Zn under conventional leaching conditions is very sensitive to the change of temperature, when the leaching temperature is increased from 60 °C to 100 °C, the leaching rate of Zn increases from 69.56 to 97.26, which may be due to the increase of temperature reduces the viscosity of ChCl-urea DESs, and the decrease of solvent surface tension accelerates the process and degree of mass transfer between liquid-solid reactants, with the increase of leaching time, the variation in Zn leaching rate tends toward equilibrium when the leaching temperature exceeds 90 °C, and the comprehensive consideration of 90 °C is the best leaching temperature under conventional conditions.

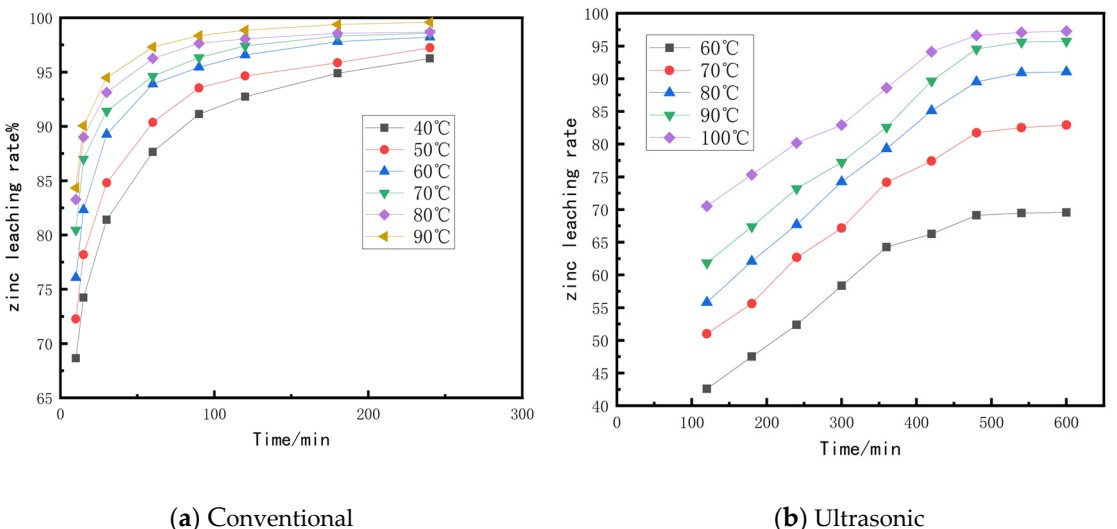

(**a**) Conventional                                           (**b**) Ultrasonic

**Figure 8.** Influence of the temperature on the zinc leaching rat (**a**) conventional: liquid-solid ratio = 16:1, stirring speed = 400 rpm; (**b**) ultrasonic: liquid-solid ratio = 15:1, ultrasonic power = 450 W.

Comparing Figure 8a and b, it can be seen that the effect of temperature on the leaching rate of Zn under ultrasonic conditions is significantly less than that under conventional conditions, the leaching rate of leaching temperature 60 °C for 90 min is the same as that of leaching temperature 90 °C for 540 min under conventional conditions, and the leaching rate of Zn reaches about 95%. Under ultrasonic conditions, the Zn leaching rate increased significantly with the increase of temperature when the leaching temperature was less than 60 °C. When the leaching temperature was higher than 60 °C, the Zn leaching rate was no longer sensitive to the change in temperature, which might be due to the fact that the increase in temperature was favorable to the ultrasonic cavitation phenomenon, when the leaching temperature was less than 60 °C, the ultrasonic cavitation effect was not obvious, and when the temperature exceeded 60 °C the system energy reached the cavitation threshold, at which time the effect of ultrasonic cavitation on the leaching

rate is greater than the effect of the water bath temperature. Therefore, under ultrasonic conditions, 60 °C is the optimal leaching temperature.

### 3.2. Effect of Liquid-to-solid Ratio on Zn Leaching Rate.

As can be seen from Figure 9, the Zn leaching rate increased significantly when the liquid-solid ratio was increased from 4:1 to 16:1 under conventional leaching conditions, and the change in leaching rate was no longer obvious when the liquid-solid ratio exceeded 16:1, which may be due to the fact that the increase in the liquid-solid ratio reduced the viscosity of the slurry and improved the diffusion conditions of the relevant ions in the reactants; under ultrasonic conditions, when the liquid-solid ratio was increased from 4:1 to 12:1, the Zn leaching rate increased the most, from 85.34 to 98.33, but the leaching rate showed a decreasing trend after the liquid-solid ratio exceeded 12:1. It may be because the generation of ultrasonic cavitation and the degree of intensification of the reaction are closely related to the solid content in the slurry, when the slurry annual decreases helps the cavitation phenomenon, but when the liquid content is too high, the bubbles generated by cavitation will increase and the cavitation intensity and efficiency will decrease. Therefore, the optimal liquid-solid ratio for ultrasonic enhanced leaching is 12:1.

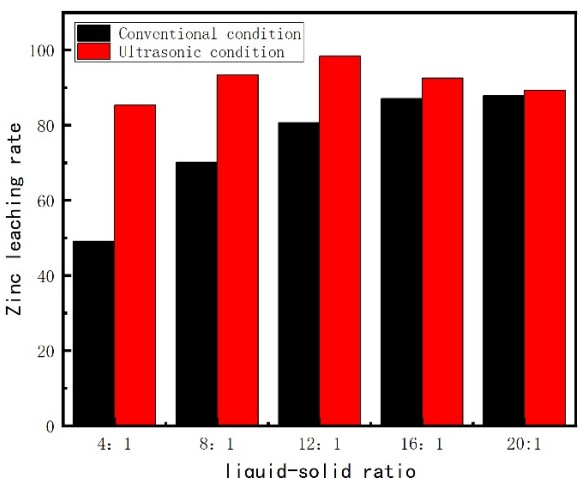

**Figure 9.** Effect of liquid-solid ratio on the zinc leaching rate; conventional: leaching temperature = 60 °C, leaching time = 240 min, stirring speed = 400 rpm, ultrasonic: leaching temperature = 60 °C, leaching time = 240 min, ultrasonic power = 450 W.

### 3.3. Effect of Time on Leaching Rate

Figure 10 shows the variation in the leaching rate of Zn with time under conventional and ultrasonic conditions, and the increase in leaching time in both cases can improve the leaching rate of Zn. The results showed that ultrasonic enhancement could significantly improve the leaching rate of Zn, which was 97.98 when the leaching time was 240 min, compared with the conventional conditions, when the leaching rate of Zn was 73.23, and the leaching rate of Zn was significantly improved by ultrasonic enhancement. In the ultrasonic enhanced leaching system, the change in Zn leaching rate was no longer obvious after the leaching time reached 240 min, while the leaching rate of Zn tended to be stable under conventional conditions until 480 min, which effectively shortened the leaching time (about 4 h) while improving the leaching rate of Zn.

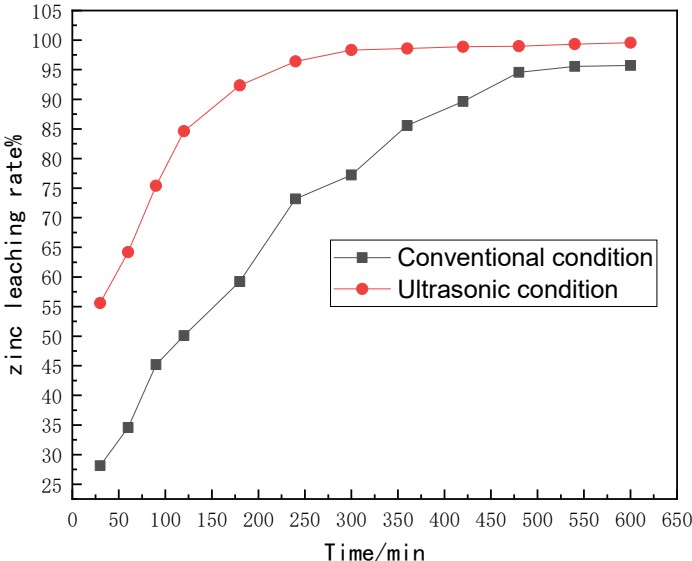

**Figure 10.** Effect of leaching time; conventional: leaching temperature = 90 °C, liquid-solid ratio =16:1, stirring speed = 400 rpm; ultrasonic: leaching temperature = 60 °C, liquid-solid ratio = 12:1, ultrasonic power = 350 W.

### 3.4. Effect of Stirring Rate and Ultrasonic Power

As shown in Figure 11a, the increase in stirring rate under conventional conditions contributes to the increase in Zn leaching rate, probably because mechanical stirring accelerates the mixing of the slurry and increases the effective surface area per unit volume of solution, which in turn increases the number of solid-liquid reaction and the reaction efficiency of the leaching agent; it can be seen from Figure 11b that when the ultrasonic power was increased from 50 W to 350 W, the leaching rate of Zn rapidly increased to 98.9%, and a large number of bubbles were generated in the slurry at this time. However, it is worth noting that the Zn leaching rate tends to decrease when the ultrasonic power exceeds 350 W. This may be because the generation and intensity of the ultrasonic cavitation effect are closely related to the ultrasonic intensity, i.e., the ultrasonic power. On the one hand, when the ultrasonic frequency is certain, the intensity of ultrasonic waves in a certain range increases, and the collapse of ultrasonic cavitation bubbles is more violent; on the other hand, when the intensity of ultrasonic waves is too large, the diameter of cavitation bubbles becomes larger, and when the cavitation bubbles are too large, the bubbles just break and do not collapse, when the ultrasonic chemical effect becomes smaller. In addition, the heat generated by too high an ultrasonic power and the local high temperature generated by the superposition of the heat outside the reaction may cause the local ChCl-urea DESs chemical property to change. Therefore, in ultrasound-enhanced leaching, it is sufficient for the ultrasonic intensity to induce a sufficiently strong cavitation effect in the slurry, and 350 W was set as the optimal ultrasonic power according to the experimental results.

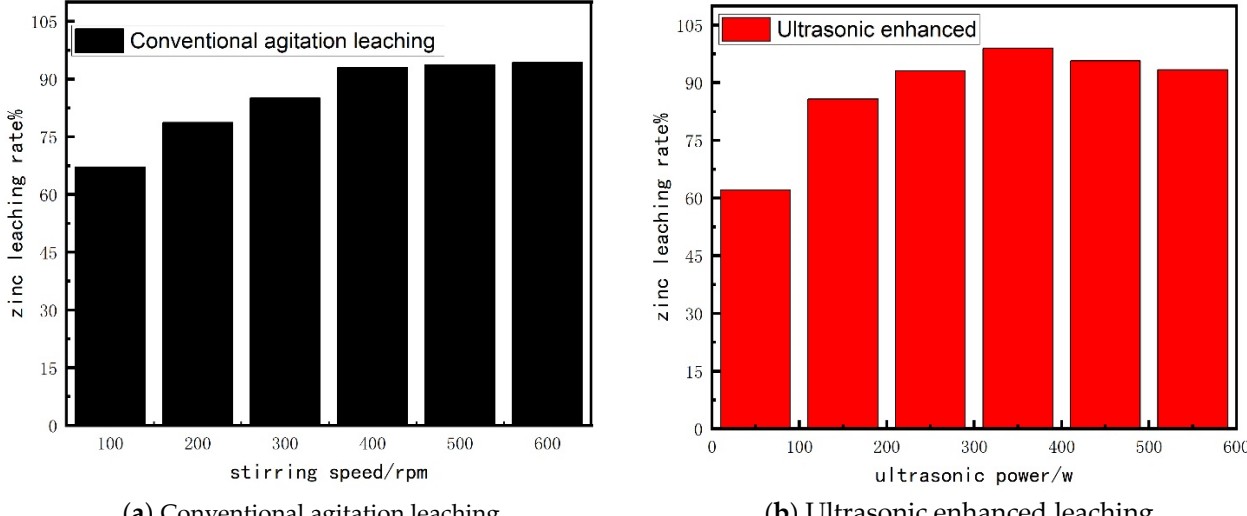

**(a)** Conventional agitation leaching　　　　**(b)** Ultrasonic enhanced leaching

**Figure 11.** effect of stirring speed and ultrasonic power on the zinc leaching rate (**a**) conventional: leaching temperature = 90 °C, , leaching time = 480 min, liquid-solid ratio = 16:1; (**b**) ultrasonic: leaching temperature e = 60 °C, , leaching time = 240 min, liquid-solid ratio = 12:1.

### 3.5. Leaching Kinetics

During the leaching process, the leaching slag of Zn-containing dust sludge was dissolved continuously, and gradually cavities and gaps appeared, the paths of leaching agent infiltration inward and complex $[ZnOCl(HOOCCH_2COOH)_2]^-$ diffusion outward grew, and the leaching rate decreased, which could provide insight into the leaching behavior of Zn-containing dust sludge from the perspective of leaching kinetics. The experimental data were substituted into Equations (2)–(4), respectively, and the fitting results are shown in Table 2.

The $R^2$ values in Table 2 reflect the degree of fit. The results show that the interfacial mass transfer and solid film layer hybrid control models fit better correlation, and the hybrid control model describes the leaching process better than other models both under conventional and ultrasonic conditions.

The relationship curves between $1/3 \ln(1-x) - 1 + (1-x)^{-1/3}$ and leaching time $t$ at different temperatures, liquid-solid ratios, stirring rates and ultrasonic powers were made according to Equation (4), respectively, as shown in Figure 12.

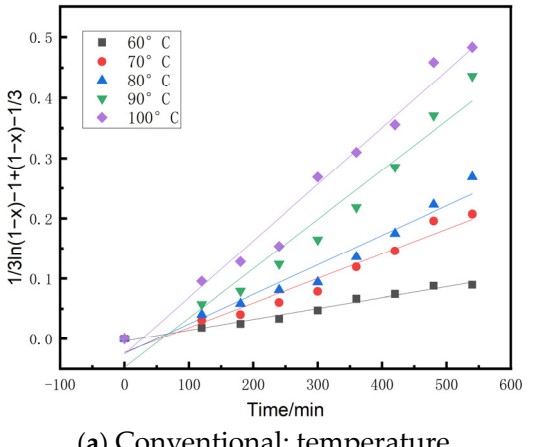

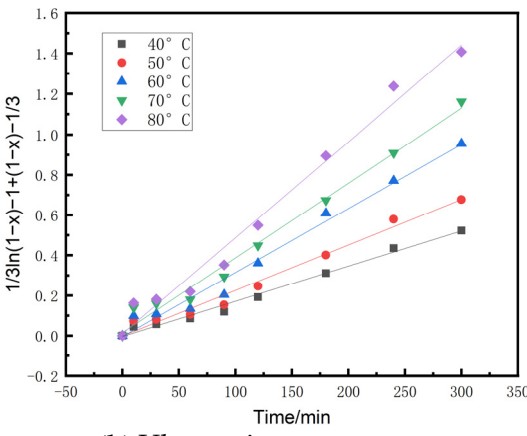

**(a)** Conventional: temperature　　　　　**(b)** Ultrasonic: temperature

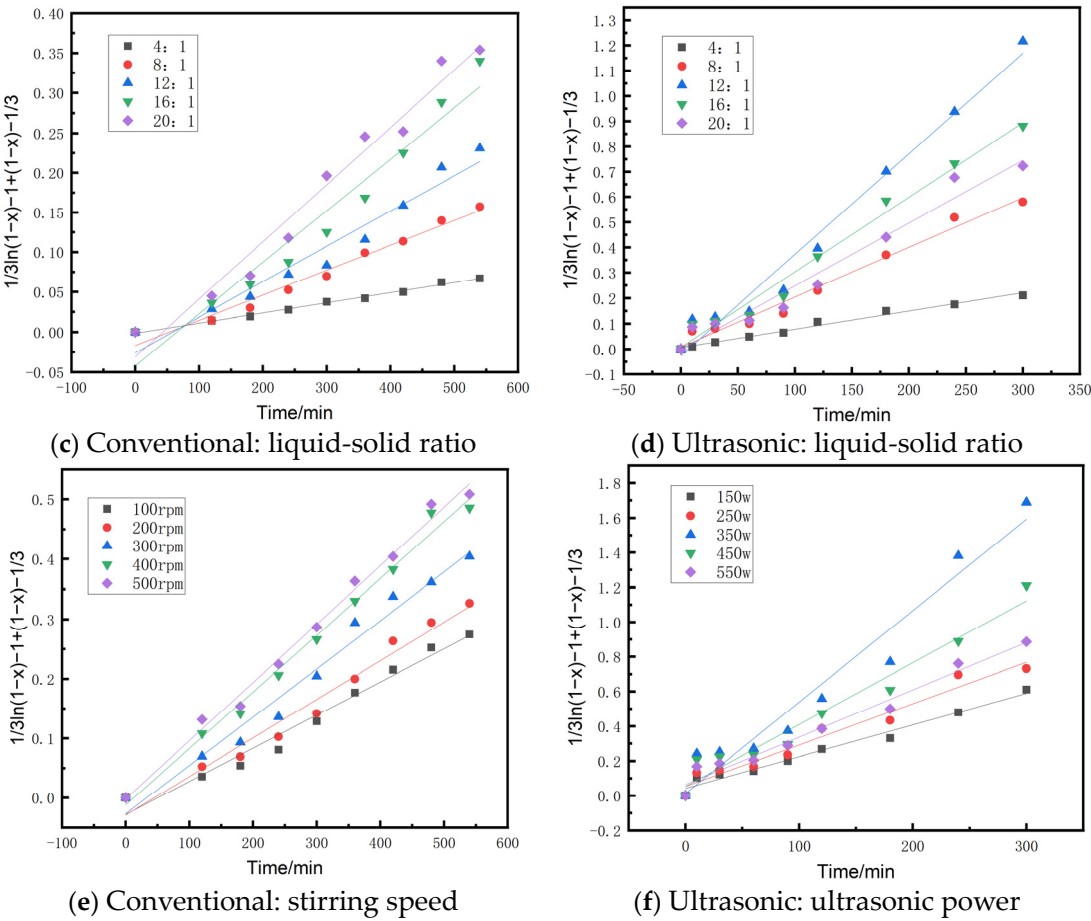

**Figure 12.** Fitting results of the "hybrid control model" at various leaching conditions

**Table 2.** Correlation coefficients R² for the three kinetic models controlled under different conditions.

| Parameters | Conditions | Solid film layer diffusion | | Surface chemical reaction | | Interfacial mass transfer and solid film layer diffusion | |
|---|---|---|---|---|---|---|---|
| | | Conventional | Ultrasonic | Conventional | Ultrasonic | Conventional | Ultrasonic |
| Liquid-solid ratio / (mL·g⁻¹) | 4:1 | 0.94913 | 0.93712 | 0.92594 | 0.87912 | 0.99041 | 0.96258 |
| | 8:1 | 0.94749 | 0.94749 | 0.93176 | 0.85761 | 0.9959 | 0.98923 |
| | 12:1 | 0.93277 | 0.92315 | 0.92056 | 0.91657 | 0.96914 | 0.99357 |
| | 16:1 | 0.90004 | 0.95781 | 0.89156 | 0.82564 | 0.96158 | 0.98954 |
| | 20:1 | 0.99759 | 0.97923 | 0.99663 | 0.79876 | 0.97744 | 0.98574 |
| Temperature /°C | 40 | 0.92052 | 0.9812 | 0.93277 | 0.89543 | 0.9784 | 0.99296 |
| | 50 | 0.88845 | 0.92763 | 0.89139 | 0.85754 | 0.96883 | 0.99553 |
| | 60 | 0.85269 | 0.87238 | 0.84417 | 0.91582 | 0.96135 | 0.99019 |
| | 70 | 0.86732 | 0.85985 | 0.85023 | 0.87351 | 0.95827 | 0.99231 |
| | 80 | 0.84273 | 0.81579 | 0.83445 | 0.91681 | 0.97799 | 0.99133 |
| Stirring speed /rpm | 100 (150) | 0.89418 | 0.78956 | 0.90533 | 0.79813 | 0.98792 | 0.98387 |
| | 200 | 0.91677 | 0.97684 | 0.92361 | 0.85436 | 0.98197 | 0.97392 |

| | | | | | | | |
|---|---|---|---|---|---|---|---|
| （Ultrasonic power /w） | （250） | | | | | | |
| | 300（350） | 0.85714 | 0.79685 | 0.84833 | 0.90563 | 0.9767 | 0.97186 |
| | 400（450） | 0.83357 | 0.91671 | 0.82916 | 0.79569 | 0.98793 | 0.99056 |
| | 500（550） | 0.90405 | 0.87964 | 0.90176 | 0.91587 | 0.98753 | 0.99095 |

The leaching data were correlated with Equation (4), and the fitted results are shown in Figure 12. The kinetics of conventional leaching and ultrasonic leaching were investigated under different conditions of temperature, liquid-to-solid ratio, stirring speed and ultrasonic power at different times. The $R^2$ values in Table 2 reflect the degree of fit, and the correlation coefficients $R^2$ of the mixed control model are all greater than 0.95, indicating that the model has a good linear relationship between the vertical coordinate and the horizontal coordinate time t. That is, the leaching process of Zn-containing dust sludge in ChCl-urea DES can be explained by this model. The reaction rate constant increases with increasing temperature, indicating that increasing the reaction temperature is beneficial to leaching, especially under conventional conditions, and the increase of the external temperature of the reaction is more obvious to the leaching rate.

The activation energy of the leaching reaction of Zn-containing dust sludge in ChCl-urea DES was calculated according to the Arrhenius equation.

$$k = Ae^{-E_a/RT} \tag{6}$$

$$\ln k = \ln A - E_a/RT \tag{7}$$

Where,

A—finger front factor.

R—gas constant, R = 8.314 $J \cdot mol^{-1} \cdot K^{-1}$.

$T$—leaching temperature, K.

$Ea$—apparent activation energy of the reaction, $kJ \cdot mol^{-1}$.

To study the reaction kinetics more clearly, the fitting results are shown in Figure 13 according to Arrhenius (7) equation with lnk as the y-axis and 1/T as the x-axis. From Equation (7), the slope of the fitting result is $-Ea/R$, and the activation energy of the reaction is calculated as 44.56 $kJ.mol^{-1}$ for the conventional leaching and 23.06 $kJ.mol^{-1}$ for the ultrasonic conditions. The activation energy results showed that the leaching rate of Zn under ultrasonic conditions was faster than that under conventional stirring conditions.

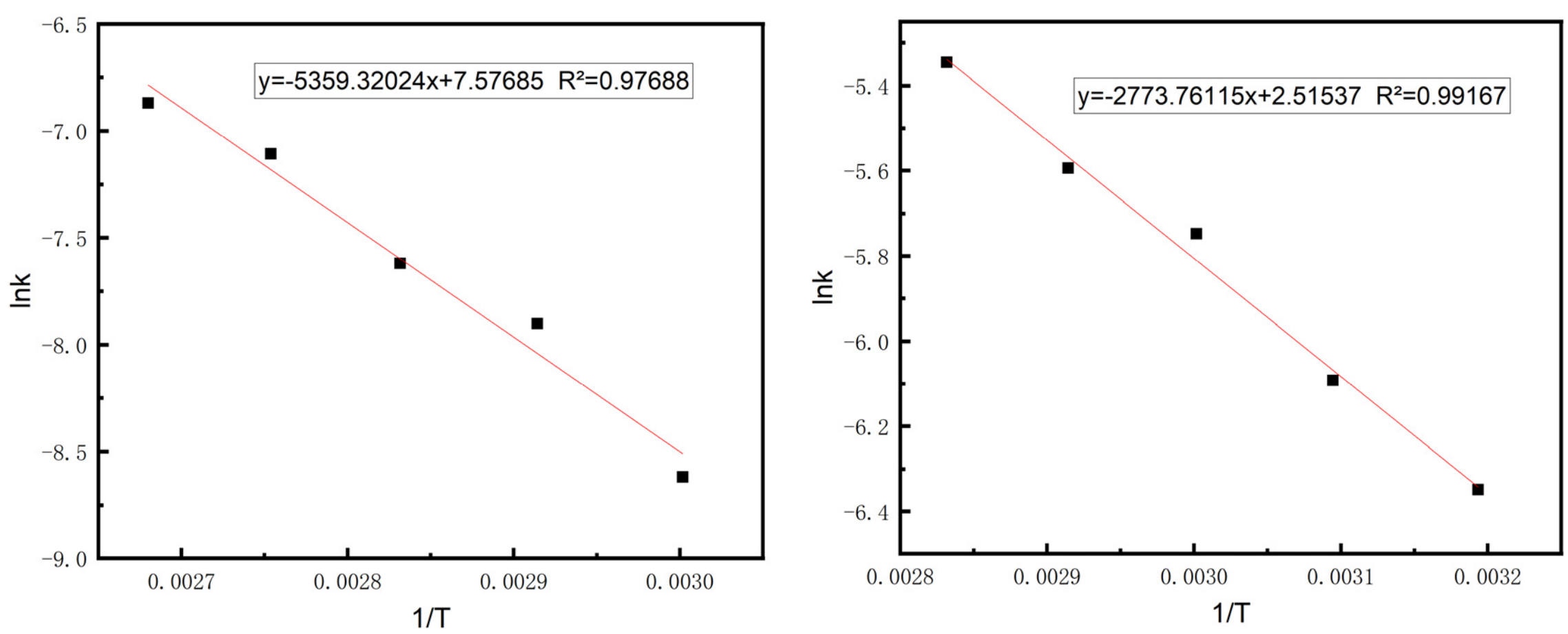


(**a**) Conventional (**b**) Ultrasonic

**Figure 13.** The relation between lnk and 1/T.

*3.6. Analysis of Leaching Residue*

Table 3 shows the chemical multi-element analysis of the leached residue under the optimum leaching conditions. As can be seen from the table, the Zn content decreased from 11.78% to 1.864% and 0.23% before and after leaching of the samples under conventional and ultrasonic leaching conditions, respectively, and most of the metallic Zn was leached out under ultrasonic conditions. The $Fe_2O_3$ content decreased from 30.67% to 25.22% and 23.36%, respectively, with leaching rates of 17% and 20%, respectively, and there was no significant change in the content of other oxides. It can be indicated that ChCl-2Urea DES has a strong solubilization ability and selectivity for metallic Zn elements in Zn-containing dust sludge.

**Table 3.** Chemical multielement analysis of leaching residue (%).

| Leaching Conditions | $Fe_2O_3$ | ZnO | $SiO_2$ | Cl | $Al_2O_3$ | CaO | PbO | MgO | $K_2O$ |
|---|---|---|---|---|---|---|---|---|---|
| Conventional | 25.220 | 1.864 | 10.182 | 2.283 | 6.176 | 3.540 | 1.497 | 1.514 | 0.155 |
| Ultrasonic | 23.36 | 0.23 | 11.63 | 0.93 | 7.173 | 4.64 | 1.56 | 1.71 | 0.19 |

As can be seen in Figure 14, the main substances in the leaching residue are hematite, magnetite, quartz and carbon. Compared with the XRD images of the zinc-containing dust sludge samples, the characteristic peaks of chlorohydric zinc ore ($Zn_5(OH)_8Cl_2$-$H_2O$) disappeared and the characteristic peaks of red zinc ore (ZnO), were significantly weakened under conventional leaching conditions, and the characteristic peaks of both chlorohydric zinc ore and red zinc ore disappeared under ultrasonic conditions, which indicated that most of the metallic zinc in the zinc-containing dust sludge was leached under ultrasonic conditions with higher leaching efficiency than conventional agitation. Simonkolleite ore was more easily dissolved in ChCl-urea DES than red zinc ore, which might be accelerated by the release of Cl- from zincSimonkolleite ore after dissolution. The peaks of magnetite become weaker and smaller under both ultrasonic and conventional conditions, which indicates that the leached metallic iron is mainly from magnetite, but the main characteristic peaks of hematite and magnetite are still present, and the peak intensity of hematite is relatively high, which indicates that ultrasonic only intensifies the leaching of metallic zinc.

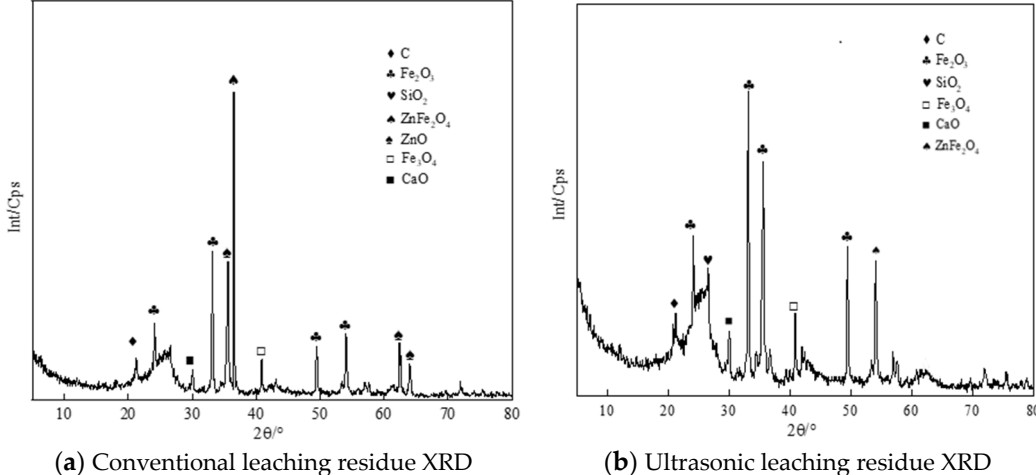

(**a**) Conventional leaching residue XRD　　　　　　　(**b**) Ultrasonic leaching residue XRD

**Figure 14.** XRD pattern of leaching residue (**a**) conventional: temperature = 60 °C, liquid-solid ratio = 12:1, stirring speed = 400 rpm, time = 480 min; (b) ultrasonic: leaching temperature = 60 °C, liquid-solid ratio = 12:1, ultrasonic power = 450W, time = 240 min.

The results of laser particle size composition analysis of the Zn-bearing dust samples are shown in Table 4. It can be found from Table 4 that the leached residue from ultrasonic leaching have a smaller particle size than that of conventional leaching. This is mainly

because the cavitation effect generated during ultrasonic enhanced leaching not only makes the mixing of solid and liquid phases more effective compared with conventional stirring leaching, but also the huge energy generated by the collapse of bubbles occurring in the cavitation effect breaks the solid particles into finer particles, which reduces the generation of reaction passivation layer and increases the reaction contact area at the same time.

**Table 4.** Leaching residue particle size composition analysis (%).

| Particle size /μm | <20 | 20~60 | 60~100 | 100~170 | >170 |
|---|---|---|---|---|---|
| Conventional | 28.21 | 28.51 | 25.74 | 13.91 | 3.63 |
| Ultrasonic | 36.10 | 32.42 | 17.72 | 11.25 | 2.51 |

Figure 15 shows the SEM microscopic morphology of the leached slag. Comparing the conventional leaching with the ultrasonic leaching, it can be found that the microstructure of the conventional leaching slag was slightly changed after 120 min of the start of the reaction, the dense bright flocculent containing zinc was reduced compared with the sample, the dense bright stripes were not changed, the particle size was larger and the distribution was disordered, and the fine particles were accompanied by coalescence. Ultrasonic leaching slag surface appeared a large number of pores and obvious grooves, dense bright-colored flocs and strips significantly reduced, and the particle size is smaller and evenly dispersed, basically no microfine particles of minerals attached, the surface is loose with voids, with no cohesion phenomenon. This indicates that when the shock wave formed by the collapse of cavitation bubbles acts on the surface of solid particles, it can generate shear stress on the surface, hinder the coalescence of insoluble materials on the surface of reaction particles, and at the same time, it can reduce the thickness of boundary layer, open the inclusions, increase the reaction contact area, reduce the solvent spreading resistance, accelerate the mass transfer process between solid and liquid phases, and facilitate the leaching reaction. Combined with the energy spectrum surface scan data, it can be found that the reaction particle inclusions mainly contain Ca, Fe, Al, Mg metal elements and Si, O, C and other non-metallic elements. In addition, the point sweep energy spectrum data in Figure 12a shows that the inclusions are mainly composed of clay minerals such as FeZn, magnetite, hematite and a small amount of limestone, and the formation of inclusions and the attachment of non-reactive materials hinder the further dissolution of Zn, while in Figure 12b, the inclusions are opened due to the ultrasonic crop and obvious peaks of Zn, C and O appear in the energy spectrum, indicating that the inclusions are opened to fully expose ZnO in the ore phase, which provides more material basis for the enhanced leaching of Zn.

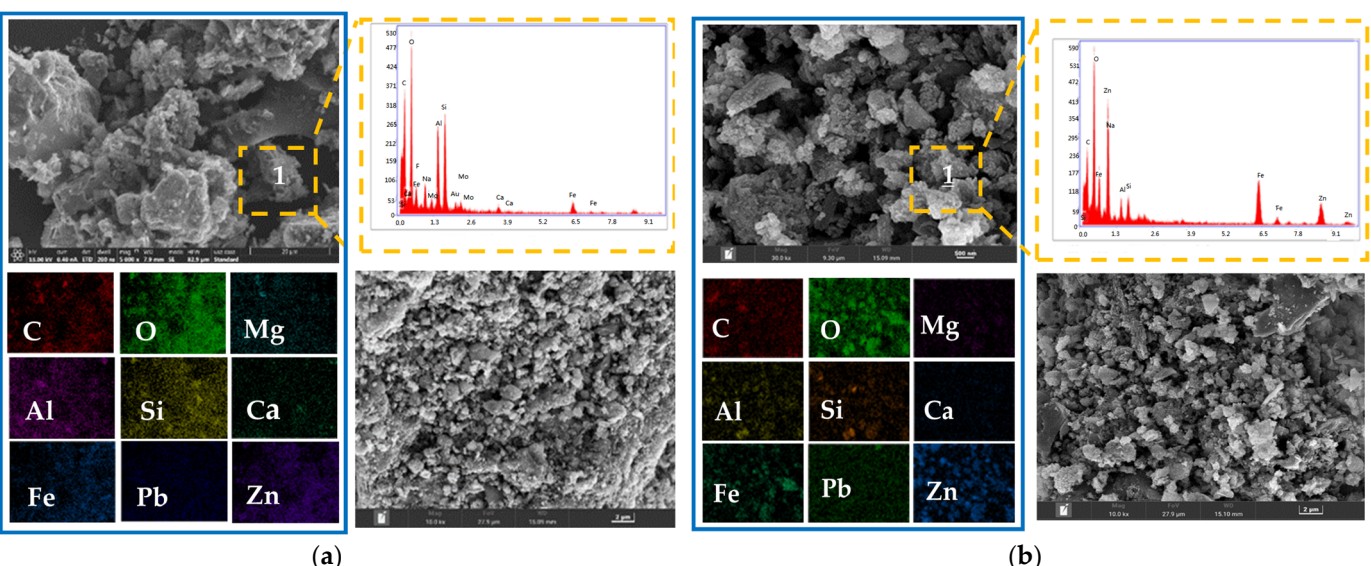

(**a**)                                                                                           (**b**)

**Figure 15.** SEM morphologies and EDS pattern of leaching residue (**a**) conventional: temperature = 60 ℃, liquid-solid ratio =12:1, stirring speed = 400 rpm, time = 120 min, (a-1) time = 240 min (**b**) ultrasonic: temperature = 60 ℃, liquid-solid ratio = 12:1, ultrasonic power = 450 W, time = 120 min (b-1) time = 240 min.

## 4. Conclusions

1.     It is feasible to use the new ionic liquid ChCl-urea DES as leaching agent to recover zinc from metallurgical Zn-containing dust sludge.

2.     Under the conditions of ultrasonic power of 350 W, liquid-solid ratio of 12:1, leaching temperature of 60 ℃ and leaching time of 240 min, the zinc leaching rate can reach 98%. Under the same conditions, ultrasonic leaching can increase the leaching rate by 30% left after the reaction time is shortened by 240 min.

3.     The kinetic results show that the reactions under both conventional and ultrasonic conditions are controlled by a mixture of interfacial mass transfer and solid film layer diffusion, with reaction activation energies of Ea1 = 44.56 kJ/mol and Ea2 = 23.06 kJ/mol, respectively.

4.     Ultrasonic waves have a great change on the microstructure of the leached slag particles. The particle boundaries of the leached slag have obvious impact traces and the particle dispersion is better than conventional conditions.

**Author Contributions:** Conceptualization, F.N. and J.Z.; methodology, J.Z and S.H.; software, J.Z.; validation, F.N. and J.Z; formal analysis, S.H and C.W.; investigation, J.Z.; data curation, S.H and C.W.; writing—original draft preparation, S.H.; writing—review and editing, F.N and J.Z.; visualization,S.H.; supervision, F.N.; project administration, J.Z.; funding acquisition, J.Z. and F.N.All authors have read and agreed to the published version of the manuscript.

**Funding:** This research was funded by the National Natural Science Foundation of China (Grant No. 51904106), the Natural Science Foundation of Hebei Province (Grant No. E2021209015), Basic Research Business Expenses for Universities in Hebei Province (Grant No. JQN2022009,)., Key projects of Hebei Provincial Department of Education(Grant No ZD2022059), Hebei Provincial High level Talents Funding Project(Grant No B20221005).

**Institutional Review Board Statement:** Not applicable

**Data Availability Statement:** The data presented in this study are available upon request from the corresponding author.

**Acknowledgments:** The authors are very grateful to the reviewers and editors for their valuable suggestions, which have helped improve the paper substantially.

**Conflicts of Interest:** The authors declare no conflict of interest

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
