# Peer review of "Study on Ultrasonically-Enhanced Deep Eutectic Solvents Leaching of Zinc from Zinc-Containing Metallurgical Dust Sludge"

_metals, doi:10.3390/met12111856_

Round 1

Reviewer 1 Report

Manuscript ID: metals-1920484

Title: Study on Ultrasonically-Enhanced Deep eutectic solvents Leaching of Zinc from Zinc-Containing Metallurgical dust sludge

Authors: Niu Fusheng et al.

Line 30-35. Authors must add information about Zn production in China for 2021 year.

Line 53-59. Authors must add information about technical parameters of acid leaching – T, concentration, L/S, metals extraction degree.

Line 65. Add full chemical formulas of choline chloride and malonic acid in brackets.

Table 1. The sum of oxides is not 100 wt. %. Why? What is the LOI and carbon (C) content?

Section 2.1. What is the particle size distribution of Zn dust?

Section 2.2. Authors must describe in detail the ultrasonic device – the capacity and the geometric shape of the sonotrode. It is better to present a scheme with geometric dimensions of the sonotrode. This is an important point, since the leaching rate depends on the shape of sonotrode.

Figure 7 and 9. Authors must add the errors bars to each column.

Table 2 and Figures 10-11 are a print screen of Word file! This is terrible! Authors need to insert the table into the file separately.

Figure 10. A very common mistake was made in the calculations. In figure 6 there is point 0, and in figure 10 the curves do not pass-through point 0, it simply is not there. This is a mistake; you need to redo the calculation. Authors can read more information here:  Chemical Reaction Engineering. https://the-seventh-dimension.com/images/textlev/LEVENSPIEL%20Chemical%20reaction%20engineering-ch1-ch2.pdf

Figure 11. Authors must sign all peaks on XRD patterns.

·        1) Authors must add information about chemical composition (wt. %) of residue after conventional and ultrasound effect.

·        2) What is the liquor chemical composition (g/L) after leaching?

·        3) How was changed the particle size distribution after leaching by conventional and ultrasound effect?

Before the conclusions, the authors should add a discussion with links about further processing of the resulting solutions.

Technical errors:

The references style and the references list is not corresponding to Metals. Authors must improve it in all article text.

Table 1. Change Al203 to Al2O3.

Line 171, 175. Use subscripts in chemical formulas.

Figure 11. XRD pattern of leaching residue and raw material must be Figure 12.

The article does not meet the level of Metals and should be rejected.

Reviewer 2 Report

In my opinion the paper "Study on ultrasonically-enhanced Deep eutectic solvents leaching of Zn from Zn-contained metallurgical dust sludge" by Niu et al cannot be accepted for publication on Metals journal. The article is not very clear and it seems poor from a methodological point of view

Comments:

Line 65. Lu instead of LU

Line 107-123. They look like results and not methods. Moreover figures 1 and 2 do not seem necessary because there is a table with the composition. The sum of the data in table 1 is 65.87, what is the remaining 35%? water?

Line 126. What is Chinese medicine? A company?

Line 127. choose between °K and °C

Line 128. why was only one molar ratio used?

Line 136-138. Remove, already written

Characterization

How many tests were performed for each experiment? Only Zn was analyzed? Is the solvent used so selective? Only Zn is removed? Has the residue been washed? How was the residue dissolved? With the same method? If so, how can we expect there to be more Zn? Some information regarding the analytical methods is necessary. what about other elements such as Co, Ni, Mn and Li reported in fig 10?

Remove fig 3

Line 155-158. How many tests for each experiment were performed?

Leaching mechanism and mathematical model

This paragraph seems redundant should it be shortened

Line 162. ZnO(001): ??

Line232-233. Subscript 0. Moreover, probably x0 it is not % but ratio (Zn/Total)

Fig 9. insert a) and b) into the graph. Why different ratio? (16:1 and 15:1)

Line 246. Remove "with the" (written twice)

Line 247. According to the figure, each curve reaches equilibrium after about 450 min

Line 255-263. It is not clear

Line 269. Probably is fig 7 and not 6

Effect of time on leaching rate

Same data of fig 6 are reported. Looking at fig 6 and comparing the data of fig 8 regarding to the conventional method there is a large difference, for example under the same conditions (450 min, 60 ° C, 400 rpm) a difference of about 25-30% is obtained. Is it possible that it is only due to the small difference in the solid-liquid ratio (16:1 vs 12:1)? Moreover, looking at fig 7 an increase in the ratio should favor leaching, it is not consistent. How many tests were done for each experiment?

Line 325. Which temperature?

Line 331-339. Already written

Table 2. Probably a multivariate analysis would better describe the impact of each parameter. It is not clear how the other parameters change for each individual test.

Fig 10. These figures refer to other elements (Li, Ni, Co and Mn) which are never mentioned in the text (and in the sludge composition). Can we, therefore, deduce that the solvent is not selective? or data coming from other not mentioned experiments?

Line 417. the instead The

Line 413-428. It is not clear

Line 423. Probably is fig 12 instead 11

Fig 12. Looking at the XRD spectra it seems that leaching removes mainly Zn5(OH)8Cl2 rather than the oxide on which the theory shown above is based.

References

Look at the instruction for authors

Reviewer 3 Report

Study on Ultrasonically-Enhanced Deep eutectic solvents Leaching of Zinc from Zinc-Containing Metallurgical dust sludge is very interesting paper. Someunprovement is required.

Line 14: Energy Dispersive Spectroscopy (EDS), X-ray diffraction (XRD)

Line 17: ultrasonic power 350w (correct: 350 W)

Line 17:  e reached (please to check it)

Line 56: to dissolve zinc ferrate, (zinc ferrite)

Line 74: 1:25 g/Ml. (1:25 g/mL)

Line 94-96: results showed that due to ultrasonic cavitation, ultrasonic waves with sulfuric acid as leaching agent could enhance the acid leaching process of potassium feldspar, and the potassium leaching rate could be increased by 3% to 96 8%. What is an explanation for this behavior (Better mixing in system?)

Line 209: the  contraction kernel model (the  contraction Kernel model)

Line 221: BINGÖ L D et al. [35]

Line 238: Conventional

Line 264: Please to put „unit for the zinc leaching rate (g/min? or %) at Figure 7!

Line 314: It can be seen from Fig. 9(b) that when the ultrasonic power was increased from 50 W to 350, the leaching rate of Zn rapidly increased to 98.9%, (Can you explain it!)

Line 317: 350 w (350 W)

Line 359:  350 w (350 W)

Line 441: .min,timen=240min (time= 240 min)

Line 452: the particle size was larger (What is particle size?)

Line 454: Ultrasonic waves have a great change on the microstructure of the leached slag particles. (Please to explain a reason for this behavior)

Round 2

Reviewer 1 Report

The authors have corrected many comments. However, the kinetics are still miscalculated.

If we analyze the points in figure 8 for ultrasound effect, we can see that there are many experimental points from 0 to 50 min. However, the authors do not use them in the calculations of the kinetics of figure 12.

This is an obvious mistake.

The kinetics analysis should use the "0" point, if there is none, the calculation is incorrect. I do not agree with the authors' explanations. The limiting stage cannot affect the method of calculating the kinetics in any way. This is the second remark.

Authors should improve section 3.5.

Round 3

Reviewer 1 Report

Article can be accepted in the present form.